# Neural Proximal Gradient Descent for Compressive Imaging

**Morteza Mardani[1], Qingyun Sun[4], Shreyas Vasawanala[2], Vardan Papyan[3],**
**Hatef Monajemi[3], John Pauly[1], and David Donoho[3]**
Depts. of [1]Electrical Eng., [2]Radiology, [3]Statistics, and [4]Mathematics; Stanford University
morteza,qysun,vasanawala,papyan,monajemi,pauly,donoho@stanford.edu

## Abstract

Recovering high-resolution images from limited sensory data typically leads to a serious ill-posed inverse problem, demanding inversion algorithms that effectively capture the prior information. Learning a good inverse mapping from training data faces severe challenges, including: (i) scarcity of training data; (ii) need for plausible reconstructions that are physically feasible; (iii) need for fast reconstruction, especially in real-time applications. We develop a successful system solving all these challenges, using as basic architecture the recurrent application of proximal gradient algorithm. We learn a proximal map that works well with real images based on residual networks. Contraction of the resulting map is analyzed, and incoherence conditions are investigated that drive the convergence of the iterates. Extensive experiments are carried out under different settings: (a) reconstructing abdominal MRI of pediatric patients from highly undersampled Fourier-space data and (b) superresolving natural face images. Our key findings include: 1. a recurrent ResNet with a single residual block unrolled from an iterative algorithm yields an effective proximal which accurately reveals MR image details. 2. Our architecture significantly outperforms conventional non-recurrent deep ResNets by 2dB SNR; it is also trained much more rapidly. 3. It outperforms state-of-the-art compressed-sensing Wavelet-based methods by 4dB SNR, with 100x speedups in reconstruction time.

## 1   Introduction

Linear inverse problems appear broadly in image restoration tasks, in applications ranging from natural image superresolution to biomedical image reconstruction. In such tasks, one oftentimes encounters a seriously ill-posed recovery task, which necessitates regularization with proper statistical priors. This is however impeded by the following challenges: c1) real-time and interactive tasks demand a low overhead for inference; e.g., imagine MRI visualization for neurosurgery [1], or, interactive superresolution on cell phones [2]; c2) the need for recovering plausible images that are consistent with the physical model; this is particularly important for medical diagnosis, which is sensitive to artifacts; c3) and limited labeled training data especially for medical imaging.

Conventional compressed sensing (CS) relies on sparse coding of images in a proper transform domain via a *universal* $\ell_1$-regularization; see e.g., [3, 4, 5]. To automate the time-intensive iterative soft-thresholding algorithm (ISTA) for sparse coding, [6] puts forth the learned ISTA (LISTA). Relying on soft-thresholding it trains a simple (single dense layer) recurrent network to map measurements to the $\ell_1$ sparse code as a surrogate for the $\ell_0$ code. [7] advocates a wider class of functions derived from proximal operators. [8] also adopts LSTMs to learn the minimal $\ell_0$ sparse code, where the learned network was seen to improve the RIP of coherent dictionaries. Sparse recovery however is the common objective of [8, 6], and the measurement model is not explicitly taken into account. No guarantees were also provided for the convergence and quality of the iterates.

Deep neural networks have recently proven quite powerful in modeling prior distributions for images [9, 10, 11, 12, 13, 14]. There is a handful of recent attempts to integrate the priors offered by generative nets for inverting linear inverse tasks dealing with local image restoration such as superresolution [10, 12], inpainting [13]; and more global tasks such as biomedical image reconstruction [15, 16, 17, 19, 20, 21, 22, 23]. One can divide them into two main categories, with the first category being the post-processing methods that train a deep network to map a poor (linear) estimate of the image to the true one [15, 17, 20, 23, 10, 12, 19]. Residual networks (ResNets) are a suitable choice for training such deep nets due to their stable training behavior [24] along with pixel-wise and perceptual costs induced e.g., by generative adversarial networks (GANs) [9, 19]. The post-processing schemes offer a clear gain in computation time, but they offer no guarantee for data fidelity. Their accuracy is also only comparable with CS-based iterative methods. The second category is inspired by unrolling the iterations of classical optimization algorithms, and learns the filters and nonlinearities by training deep CNNs [25, 16, 26, 27]. They improve the accuracy relative to CS, but deep denoising CNNs that are changing over iterations incur a huge training overhead. Note also that for a signal that has a low-dimensional code under a deep pre-trained generative model, [28, 29] establishes reconstruction guarantees. The inference however relies on a iterative procedure based on empirical risk minimization that is quite time intensive for real-time applications. **Contributions.** Aiming for rapid, feasible, and plausible image recovery in ill-posed linear inverse tasks, this paper puts forth a novel neural proximal gradient descent algorithm that learns the proximal map using a recurrent ResNet. Local convergence of the iterates is studied for the inference phase assuming that the true image is a fixed point for a proximal (lies on a manifold represented by proximal). In particular, contraction of the learned proximal is empirically analyzed to ensure the RNN iterates converge to the true solution. Extensive evaluations are examined for the global task of MRI reconstruction, and a local task of natural image superresolution. We find:

- For MRI reconstruction, it works better to repeat a small ResNet (with a single RB) several times than to build a general deep network.

- Our recurrent ResNet architecture outperforms general deep network schemes by about 2dB SNR, with much less training data needed. It is also trained much more rapidly.

- Our architecture outperforms existing state-of-the-art CS-WV schemes, with a 4dB gain in SNR, while achieving reconstruction with 100x reduction in computing time.

These findings rest on several novel project contributions:

- Successful design and construction of a neural proximal gradient descent scheme based on recurrent ResNets.

- Rigorous experimental evaluations, both for undersampled pediatric MRI data, and for superresolving natural face images, comparing our proposed architecture with conventional non-recurrent deep ResNets and with CS-WV.

- Formal analysis of the map contraction for the proximal gradient algorithm with accompanying empirical measurements.

## 2 Preliminaries and problem statement

Consider an ill-posed linear system $y = \Phi x_* + v$ with $\Phi \in \mathbb{C}^{m \times n}$ where $m \ll n$, and $v$ captures the noise and unmodeled dynamics. Suppose the unknown and (complex-valued) image $x$ lies in a *low-dimensional* manifold. No information is known about the manifold besides the training samples $\mathcal{X} := \{x_i\}_{i=1}^N$ drawn from it, and the corresponding (possibly) noisy observations $\mathcal{Y} := \{y_i\}_{i=1}^N$. Given a new undersampled observation $y$, the goal is to *quickly* recover a plausible image $x_*$.

The stated problem covers a wide range of image restoration tasks. For instance, in medical image reconstruction, $\Phi$ describes a projection driven by physics of the acquisition system (e.g., Fourier transform for MRI scanner, and Radon transform for the CT scanner). For image superresolution it is the downsampling operator that averages out nonoverlapping image regions to arrive at a low-resolution image. Given an image prior distribution, one typically forms a maximum-likelihood estimator formulated as a regularized least-squares (LS) program

$$(\text{P1}) \qquad \min_x \ \frac{1}{2}\big\|y - \Phi x\big\|^2 + \psi(x; \mathcal{W}) \tag{1}$$

with the regularizer $\psi(\cdot)$ parameterized by $\mathcal{W}$ that incorporates the image prior.

In order to solve (P1) one can adopt a variation of proximal gradient algorithm [30] with a proximal operator $\mathcal{P}_\psi$ that depends on the regularizer $\psi(\cdot, \cdot)$ [30]. Starting from $x_0$, and adopting a small step size $\alpha$ the overall iterative procedure is expressed as

$$x_{t+1} = \mathcal{P}_\psi\left(x_t - \alpha\nabla\frac{1}{2}\|y - \Phi x_t\|^2\right) = \mathcal{P}_\psi\left(x_t + \alpha\Phi^{\mathsf{H}}(y - \Phi x_t)\right) \quad (2)$$

For convex function $\psi$, the proximal map is monotone, and the fixed point of (2) coincides with the global optimum for (P1) [30]. For some simple prior distributions, the proximal operation is tractable in closed-form. One popular example of such a proximal pertains to $\ell_1$-norm regularization for sparse coding, where the proximal operator gives rise to soft-thresholding and shrinkage in a certain domain such as Wavelet, or, Fourier. The associated iterations have been labeled ISTA; the related FISTA iterations offer accelerated convergence [31].

# 3 Neural Proximal learning

Motivated by the proximal gradient iterations in (2), to design efficient network architectures that automatically invert linear inverse tasks, the following questions need to be first addressed:

*Q1. How to ensure rapid inference with affordable training for real-time image recovery?*

*Q2. How to ensure plausible reconstructions that are physically feasible?*

## 3.1 Deep recurrent network architecture

The recursion in (2) can be envisioned as a feedback loop which at the $t$-th iteration takes an image estimate $x_t$, moves it towards the affine subspace of data consistent images, and then applies the proximal operator to obtain $x_{t+1}$. The iterations adhere to the state-space model

$$s_{t+1} = g(x_t; y) \quad (3)$$
$$x_{t+1} = \mathcal{P}_\psi(s_{t+1}) \quad (4)$$

where $g(x_t; y) := \alpha\Phi^{\mathsf{H}}y + (I - \alpha\Phi^{\mathsf{H}}\Phi)x_t$ is the gradient descent step that encourages data consistency. The initial input is $x_0 = 0$, with initial state $s_1 = \alpha\Phi^{\mathsf{H}}y$ that is a linear (low-quality) image estimate. The state variable is essentially a linear network with the learnable step size $\alpha$ that linearly combines the linear image estimate $\Phi^{\mathsf{H}}y$ with the output of the previous iteration, namely $x_t$.

In order to model the proximal mapping we use a homogeneous recurrent neural network depicted in Fig. 1. In essence, a truncated RNN with $T$ iterations is used for training. The measurement $y$ forms the input variables for all iterations, which together with the output of the previous iteration form the state variable for the current iteration. The proximal operator is modeled via a possibly deep neural network, as will be elaborated in the next section. As argued earlier, the proximal resembles projection onto the manifold of visually plausible images. Thus, one can interpret $\mathcal{P}_\psi$ as a denoiser that gradually removes the aliasing artifacts from the input image.

## 3.2 Proximal modeling

We consider a $K$-layer neural network with element-wise activation function $\sigma(z) = D(z) \cdot z$. We study several examples of the mask function $D(z)$, including the step function for ReLU, and the sigmoid function for Swish [32]. The $k$-th layer maps $h_{k-1}$ to $h_k$ through

$$z_k = W_k h_{k-1},$$
$$h_k = \sigma(z_k) = D(z_k) \cdot z_k$$

where the bias term is included in the weight matrix. At the $t$-th iteration, the network starts with the input $z_0 = x_t$, and outputs $z_K := x_{t+1}$. Typically, the linear weights $W_k$ are modeled by a convolution operation with a certain kernel and stride size. The network weights collected in $\mathcal{W} := \{W_k\}_{k=1}^K$ then parameterize the proximal. To avoid vanishing gradients associated with training RNNs we can use ResNets [24] or, highway nets [33]. An alternate path to our model goes via DiracNets [34] with $W_k = I + \bar{W}_k$, which are shown to exhibit similar behavior as ResNet.

## 3.3 Neural proximal training

In order to learn the proximal map, the recurrent neural network in Fig. 1 is trained end-to-end using the population of training data $\mathcal{X}$ and $\mathcal{Y}$. For the measurement $y_i$, RNN with $T$ iterations recovers

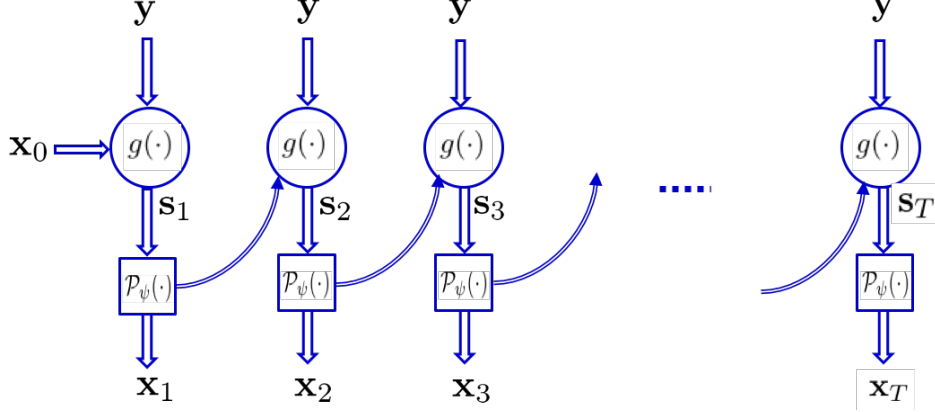

Figure 1: Truncated RNN architecture for neural proximal learning with $T$ iterations. $\mathcal{P}_\psi$ is modeled with a multi-layer NN.

$\hat{x}_i = x_T^i = (\mathcal{P}_\psi \circ g)^T(\Phi^{\mathsf{H}} y_i)$, where the composite map $\mathcal{P}_\psi \circ g$ is parameterized by the network training weights $\mathcal{W}$ and step size $\alpha$. Let $\hat{\mathcal{X}}$ denote the population of recovered images. In general, one can use the population-wise costs such as GANs [9, 19], or, the element-wise costs such as $\ell_1/\ell_2$ to penalize the difference between $\mathcal{X}$ and $\hat{\mathcal{X}}$. To ease the exposition, we adopt the element-wise empirical risk minimization

$$(\text{P2}) \quad \min_{\mathcal{W},\alpha} \; \beta \sum_{i=1}^{N} \ell\big(x_i, x_i^T\big) + (1-\beta)\sum_{i=1}^{N}\sum_{t=1}^{T} \|y_i - \Phi x_i^t\|^2$$
$$\text{s.t.} \quad \hat{x}_i^t = (\mathcal{P}_\psi \circ g)^t(\Phi^{\mathsf{H}} y_i), \quad \forall i \in [N], \; t \in [T]$$

for some $\beta \in [0, 1]$, where a typical choice for loss $\ell$ is MSE, i.e., $\ell(\hat{x}, x) = \|x - \hat{x}\|^2$. The second term encourages the outputs of different iterations to be consistent with the measurements. It is found to significantly improve training convergence of RNN for large iteration numbers $T$ when the gradient vanishing can occur. Alternatively, to facilitate the training one can ask reconstructions at each iteration to be faithful with the ground-truth images as in [18]. Note, one can additionally augment (P2) with adversarial GAN loss as in our companion work [19] that favors more the image perceptual quality that is critical in medical imaging.

## 4  Contraction Analysis

Consider the trained RNN in Fig. 1. In the inference phase with a new measurement $y$, we are motivated to study whether the iterates $\{(s_t, x_t)\}$ in (3)-(4) converge, their speed of convergence, and whether upon convergence they coincide with the true unknown image. To make the analysis tractable, the following assumptions are made:

*(A1)* The measurements are noiseless, namely, $y = \Phi x_*$. and the true image $x_*$ is close to a fixed point of the proximal operator, namely $\|x_* - \mathcal{P}_\psi(x_*)\| \le \epsilon$ for some small $\epsilon$.

The fixed point assumption seems to be an stringent requirement, but it is typically made in this context to make the analysis tractable; see e.g., [28]. It roughly means that the images lie on a manifold [1] represented by the map $\mathcal{P}_\psi$. Assuming that the train and test data lie on the same manifold, one can enforce it during the training by adding a penalty term to (P2).

The mask can then be decomposed as

$$d_t^k = D(z_*^k) + (D(z_t^k) - D(z_*^k)) = d_*^k + \delta_t^k. \tag{5}$$

where $d_*^k = D(z_*^k)$ is the true mask, and $\delta_t^k$ models the perturbation. Passing the input image $x_t$ into the $K$-layer neural network then yields the output

$$x_{t+1} = M_t^K \ldots M_t^2 M_t^1(\alpha \Phi^{\mathsf{H}} y + (I - \alpha \Phi^{\mathsf{H}} \Phi) x_t), \tag{6}$$

where $M_t^k = \text{diag}(d_t^k)W^k$. One can further write $M_t^k$ as

$$M_t^k = \text{diag}(d_*^k + \delta_t^k)W^k = \text{diag}(d_*^k)W^k + \text{diag}(\delta_t^k)W^k = M_*^k + \text{diag}(\delta_t^k)W^k. \tag{7}$$

Let us define the residual operator

$$\Delta_t := \underbrace{M_t^K \ldots M_t^2 M_t^1}_{:=M_t} - \underbrace{M_*^K \ldots M_*^2 M_*^1}_{:=M_*}. \tag{8}$$

It can then be expressed as $\Delta_t = \Delta_t^1 + \ldots + \Delta_t^K$ with

$$\Delta_t^s := \sum_{j_1,\ldots,j_s} M_*^K \ldots (\text{diag}(\delta_t^{j_s})W^{j_s}) \ldots (\text{diag}(\delta_t^{j_1})W^{j_1}) \ldots M_*^1. \tag{9}$$

The term $\Delta_t^s$ captures the mask perturbation in every $s$-subset of the layers.

Rearranging the terms in (6), and using the assumption (A1), namely $M_* x_* = x_* + \xi$ for some representation error $\xi$ such that $\|\xi\| \leq \epsilon$, and the noiseless model $y = \Phi x_*$, we arrive at

$$
\begin{aligned}
x_{t+1} - x_* &= (M_* + \Delta_t)(\alpha \Phi^{\mathsf{H}}\Phi x_* + (I - \alpha\Phi^{\mathsf{H}}\Phi)x_t) - x_* \\
&= M_*(I - \alpha\Phi^{\mathsf{H}}\Phi)(x_t - x_*) + \Delta_t(I - \alpha\Phi^{\mathsf{H}}\Phi)(x_t - x_*) + \Delta_t x_* + \xi
\end{aligned}
\tag{10}
$$

To study the contraction property and thus local convergence of the iterates $\{x_t\}$ to the true solution $x_*$, let us first suppose that the perturbation $x_t - x_*$ at $t$-th iteration belongs to the set $\mathcal{S}_t$. We then introduce the contraction parameter associated with $M_*$ as

$$\eta_1^t := \sup_{\delta \in \mathcal{S}_t} \frac{\|M_*(I - \alpha\Phi^{\mathsf{H}}\Phi)\delta\|}{\|\delta\|}. \tag{11}$$

Similarly, for the perturbation map $\Delta_t$ define the contraction parameter

$$\eta_2^t := \sup_{\delta \in \mathcal{S}_t} \frac{\|\Delta_t[x_* + (I - \alpha\Phi^{\mathsf{H}}\Phi)\delta]\|}{\|\delta\|} \tag{12}$$

Applying triangle inequality to (10), one then simply arrives at

$$\|x_{t+1} - x_*\| \leq \|M_*(I - \alpha\Phi^{\mathsf{H}}\Phi)(x_t - x_*)\| + \|\Delta_t[(I - \alpha\Phi^{\mathsf{H}}\Phi)(x_t - x_*) + x_*]\| + \|\xi\| \tag{13}$$

$$\leq (\eta_1^t + \eta_2^t)\|x_t - x_*\| + \epsilon \tag{14}$$

According to (14), for small values $\epsilon \approx 0$ a sufficient condition for (asymptotic) linear convergence of the iterates $\{x_t\}$ to true $x_*$ is that $\limsup_{t\to\infty}(\eta_1^t + \eta_2^t) < 1$. For the non-negligible representation error $\xi$, if one wants the iterates to converge within a $\nu$-ball of $x_*$, i.e., $\|x_t - x_*\| \leq \nu$, a sufficient condition is that $\limsup_{t\to\infty}(\eta_1^t + \eta_2^t) < 1 - \epsilon/\nu$.

Motivated by real-time applications, e.g., in MRI neurosurgery visualization, it is of high interest to use the minimum iteration count $T$ that algorithm reaches within a close neighborhood of $x_*$. Our conjecture is that for a reasonably expressive neural proximal network, the perturbation masks $\delta_t^j$ become highly sparse for the perturbed layers over the iterations so as $\eta_2^t \leq \epsilon_2$, $t \geq T$ for some small $\epsilon_2$. Further analysis of this phenomenon, and establishing guarantees under simple and interpretable conditions in terms of network parameters is an important next step. This is the subject of our ongoing research, and will be reported elsewhere. Nonetheless, the next section provides empirical observations about the contraction parameters, where in particular $\eta_1^t$ is observed to be an order-of-magnitude larger than $\eta_2^t$.

**Remark 1 [De-biasing].** In sparse linear regression, LASSO is used to obtain a sparse solution that is possibly biased, while the support is accurate. The solution can then be de-biased by solving a LS program given the LASSO support. In a similar manner, neural proximal gradient descent may introduce a bias due to e.g., the representation error $\xi$. To reduce the bias, after the convergence of iterates to $x_T$, one can fix the masks at all layers and replace the proximal map with the linear map $M_T$, and then find another fixed point for the iterates (6).

# 5 Experiments

Performance of our novel neural proximal gradient descent scheme was assessed in two tasks: reconstructing pediatric MR images from undersampled k-space data; and superresolving natural face images. In the first task, undersampling k-space introduces aliasing artifacts that globally impact the entire image, while in the second task the blurring is local. While our focus is mostly on MRI, experiments with the image superresolution task are included to shed some light on the contraction analysis in previous section. In particular, we aim to address the following questions:

*Q1. What is the performance compared with the conventional deep architectures and with CS-MRI?*

*Q2. What is the proper depth for the proximal network, and number of iterations (T) for training?*

*Q3. Can one empirically verify the deep contraction conditions for the convergence of the iterates?*

## 5.1 ResNets for proximal training

To address the above questions, we adopted a ResNet with a variable number of residual blocks (RB). Each RB consisted of two convolutional layers with $3 \times 3$ kernels and a fixed number of $128$ feature maps, respectively, that were followed by batch normalization (BN) and ReLU activation. We followed these by three simple convolutional layers with $1 \times 1$ kernels, where the first two layers undergo ReLU activation.

We used the Adam SGD optimizer with the momentum parameter $0.9$, mini-batch size 2, and initial learning rate $10^{-5}$ that is halved every 10K iterations. Training was performed with TensorFlow interface on an NVIDIA Titan X Pascal GPU with 12GB RAM. The source code for TensorFlow implementation is publicly available in the Github page [35].

## 5.2 MRI reconstruction and artifact suppression

Performance of our novel recurrent scheme was assessed in removing k-space undersampling artifacts from MR images. In essence, the MR scanner acquires Fourier coefficients (k-space data) of the underlying image across various coils. We focused on a single-coil MR acquisition model, where for the $n$-th patient, the acquired k-space data admits

$$y_{i,j}^{(n)} = [\mathcal{F}(x_n)]_{i,j} + v_{i,j}^{(n)}, \ \ (i,j) \in \Omega \tag{15}$$

Here, $\mathcal{F}$ refers to the 2D Fourier transform, and the set $\Omega$ indexes the sampled Fourier coefficients. Just as in conventional CS MRI, we selected $\Omega$ based on variable-density sampling with radial view ordering that is more likely to pick low frequency components from the center of k-space [4]. Only $20\%$ of Fourier coefficients were collected.

**Dataset**. T1-weighted abdominal image volumes were acquired for $350$ pediatric patients. Each 3D volume includes $151$ axial slices of size $200 \times 100$ pixels. All in-vivo scans were acquired on a 3T MRI scanner (GE MR750) with voxel resolution $1.07 \times 1.12 \times 2.4$ mm. The input and output were complex-valued images of the same size and each included two channels for real and imaginary components. The input image was generated using an inverse 2D FT of the k-space data where the missing data were filled with zeros (ZF); it is severely contaminated with artifacts.

### 5.2.1 Performance for various number/size of iterations

In order to assess the impact of network architecture on image recovery performance, the RNN was trained for a variable number of iterations ($T$) with a variable number of residual blocks (RBs). 10K slices (67 patients) from the train dataset were randomly picked for training, and $1,280$ slices (9 patients) from the test dataset for test. For training RNN, we use $\ell_2$ cost in (P2) with $\beta = 0.75$.

Fig. 2 depicts the SNR and structural similarity index metric (SSIM) [36] versus the number of iterations (copies), when proximal network comprises $1/2/5/10$ RBs. It is observed that increasing the number of iterations significantly improves the SNR and SSIM, but lead to a longer inference and training time. In particular, using three iterations instead of one achieves more than 2dB SNR gain for 1 RB, and more than 3dB for 2 RBs. Interestingly, when using a single iteration, adding more than 5 RBs to make a deeper network does not yield further improvements; the SNR=24.33 for 10 RBs, and SNR=24.15 for 5 RBs. Notice also that a single RB tends to be reasonably expressive to model the MR image denoising proximal, and as a result, repeating it several times, the SNR does not seem to exceed 27dB. Using 2 RBs however turns out to be more expressive to learn the proximal, and perform as good as using 5 RBs. Similar observations are made for SSIM.

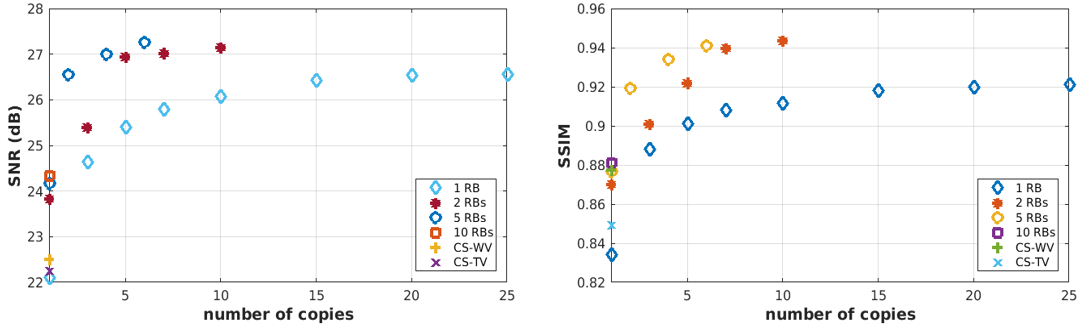

Figure 2: Average SNR and SSIM versus the number of copies (iterations). Note, single copy ResNet refers to the deep ResNet that is an exiting alternative to our proposed RNN.

Table 1: Performance trade-off for various RNN architectures.

| iterations | RBs | train time (hours) | inference time (sec) | SNR (dB) | SSIM |
|---|---|---|---|---|---|
| 10 | 1 | 2 | 0.04 | 26.07 | 0.9117 |
| 5 | 2 | 4 | 0.10 | 26.94 | 0.9221 |
| 2 | 5 | 8 | 0.12 | 26.55 | 0.9194 |
| deep ResNet | 10 | 12 | 0.0522 | 24.33 | 0.8810 |
| CS-TV | n/a | n/a | 1.30 | 22.20 | 0.82 |
| CS-WV | n/a | n/a | 1.16 | 22.51 | 0.86 |

**Training and inference time.** Inference time is proportional to the number of unrolled iterations. Passing each image through one unrolled iteration with one RB takes 4 msec when fully using the GPU. It is hard to precisely evaluate the training and inference time under fair conditions as it strongly depends on the implementation and the allocated memory and processing power per run. Estimated inference times as listed in Table 1 are averaged out over a few runs on the GPU. We observed empirically that with shared weights, e.g., 10 iterations with 1 RB, the training converges in $2 - 3$ hours. In constrast, training a deep ResNet with 10 RBs takes around $10 - 12$ hours to converge.

### 5.2.2 Comparison with sparse coding

To compare with conventional CS-MRI, CS-WV is tuned for best SNR performance using BART [37] that runs 300 iterations of FISTA along with 100 iterations of conjugate gradient descent to reach convergence. Quantitative results are listed under Table 1, where it is evident that the recurrent scheme with shared weights significantly outperforms CS with more than 4dB SNR gain that leads to sharper images with finer texture details as seen in Fig. 3. As a representative example, Fig. 3 depicts the reconstructed abdominal slice of a test patient. CS-WV retrieves a blurry image that misses out the sharp details of the liver vessels. A deep ResNet with one iteration and 10 RBs captures a cleaner image, but still blurs out fine texture details such as vessels. However, when using 10 unrolled iterations with a single RB for proximal modeling, more details of the liver vessels are visible, and the texture appears to be more realistic. Similarly, using 5 iterations and 2 RBs retrieves finer details than 2 iterations with relatively large 5 RBs network for proximal.

In summary, we make three key findings:

**F1.** The proximal for denoising MR images can be well represented by training a ResNet with a small number $1 - 2$ of RBs.

**F2.** Multiple back-and-forth iterations are needed to recover a plausible MR image that is physically feasible.

**F3.** Considering the training and inference overhead and the quality of reconstructed images, RNN with 10 iterations and 1 RB proximal is promising to implement in clinical scanners.

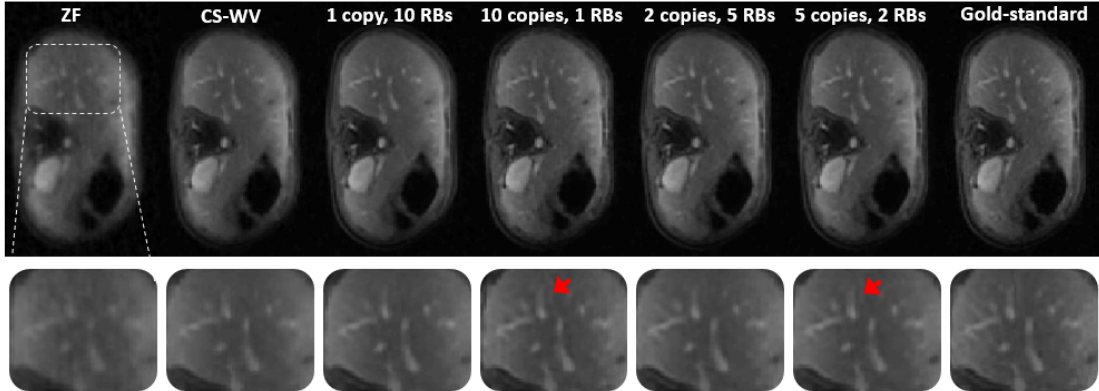

Figure 3: A representative axial abdominal slice for a test patient reconstructed by zero-filling (1st column); CS-WV (2nd column); deep ResNet with 10 RBs (3rd column); and neural proximal gradient descent with 10 iterations and 1 RBs (4th column), 2 iterations and 5 RBs (5th column), 5 iterations and 2 RBs (6th column); and the gold-standard (7th column).

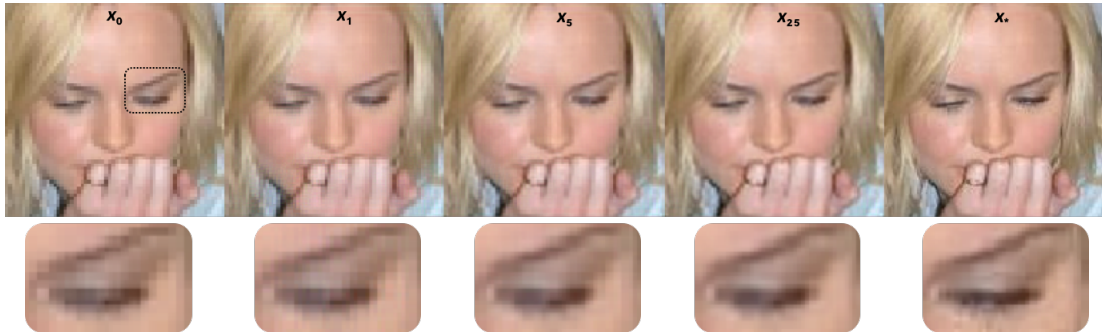

Figure 4: Superresolved ($2\times$) face images at different iterations ($x_0, x_1, x_5, x_{25}$) compared with the ground-truth ($x_*$). Proximal is a single nonlinear layer CNN with kernel size 32.

## 5.3 Verification of the contraction conditions

To verify the contraction analysis developed for Proposition 1, we focus on the image superresolution (SR) task. In this linear inverse task, one only has access to a low-resolution (LR) image $y = \phi * x$ downsampled via the convolution kernel $\phi$. To form $y$, the image pixels in $2 \times 2$ non-overlapping regions are averaged out. SR is a challenging ill-posed problem, and has been subject of intensive research; see e.g., [38, 39, 10, 2]. Our goal is not to achieve state-of-the-art performance, but a simple scenario to study the behavior of contraction parameters for proximal learning.

**CelebA dataset.** Adopting celebFaces Attributes Dataset (CelebA) [40], for training and test we use 10K and 1,280 images, respectively. Ground-truth images has $128 \times 128$ pixels that is downsampled to $64 \times 64$ LR images.

The proximal net is modeled as a 5-layer linear CNN with Smash nonlinearity [32] in the last layer. The hidden layers undergo no nonlinearity and the kernel size 8 and 32 are adopted. Thus, it is effectively a single layer nonlinear neural network. The proximal then admits $\mathcal{P}_\psi(x) = \sigma(Wx)$ as per (6). RNN with $T = 25$ is trained, and normalized RMSE, i.e., $\|x_t - x_*\|/\|x_*\|$ is plotted versus the iteration index in Fig. 5 (top) for various kernel sizes. It decreases quickly and after a few iterations it converges which suggests that the converged solution is possibly a fixed point for the proximal map. For a representative face image, output of different iterations $t_0, t_1, t_5, t_{25}$ as well as the ground-truth $x_*$ are plotted in Fig. 4. Apparently, the resolution improves over the iterations.

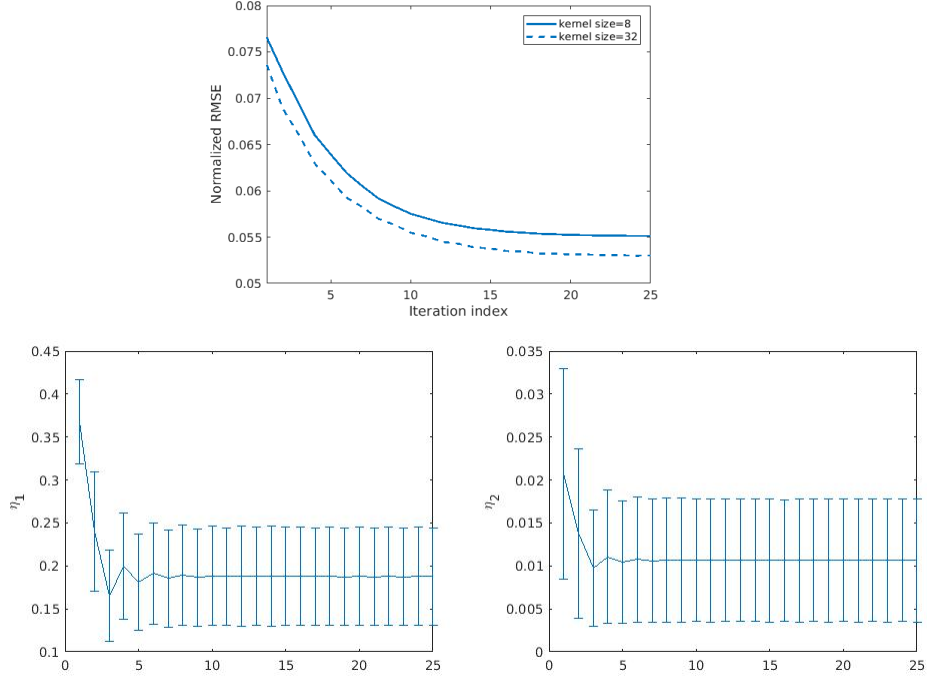

Figure 5: The top figure is normalized RMSE evolution over iterations for image superresolution task with different kernel sizes. The bottom ones are also the error bar for $\eta_1$ and $\eta_2$ per iteration for image superresolution where the proximal is a single nonlinear layer CNN.

The contraction parameters are also plotted in Fig. 5. The space of perturbations for the operator norm are limited to the admissible ones that inherit the structure of iterations. For the $i$-th test sample, we inspect the behavior $\eta_{1,t}^i = \|M^*(I - \alpha\Phi^H\Phi)\delta_t^i\|/\|\delta_t^i\|$, where $\delta_t^i := x_t^i - x_*^i$. The corresponding error bars are then plotted in Fig. 5 for kernel size 32. It is apparent that $\eta_{1,t}^i$ and $\eta_{2,t}^i$ quickly decay across iterations, indicating that later iterations produce perturbations that are more incoherent to the proximal map. Also, we can see that $\eta_{2,t}^i$ converges to a level that represents the bias generated by the iterates, similar to the bias introduced in LASSO. In addition, one can observe that $\eta_{1,t}^i$ is the dominant term - usually an order of magnitude larger than $\eta_{2,t}^i$.

## 6 Conclusions

This paper develops a novel neural proximal gradient descent scheme for recovery of images from highly compressed measurements. Unrolling the proximal gradient iterations, a recurrent architecture is proposed that models the proximal map via ResNets. For the trained network, contraction of the proximal map and subsequently the local convergence of the iterates is studied and empirically evaluated. Extensive experiments are performed to assess various network wirings, and to verify the contraction conditions in reconstructing MR images of pediatric patients, and superresolving natural images. Our findings for MRI indicate that a small ResNet can effectively model the proximal, and significantly improve the quality and complexity of recent deep architectures as well as conventional CS-MRI.

While this paper sheds some light on the local convergence of neural proximal gradient descent, our ongoing research focuses on a more rigorous analysis to derive simple and interpretable contraction conditions. The main challenge pertains to understanding the distribution of activation masks that needs extensive empirical evaluation. other important avenues that are the focus of our current research include: 1)Stable training of neural PGD for large iteration counts using gated recurrent networks; 2) comparing with existing deep learning based MRI reconstruction schemes such as deep ADMM-net and LDMAP; 3) more extensive experiments for natural image superresolution with deeper proximals and possibly using dilated convolutions for capturing large image field of view.

## 7 Acknowledgements

We would like to acknowledge Dr. Marcus Alley from the Radiology Department at Stanford University for setting up the infrastructure to automatically collect the MRI dataset used in this paper. We would also like to acknowledge Dr. Enhao Gong, and Dr. Joseph Cheng for fruitful discussions and their feedback about the MRI reconstruction and software implementation.

## Footnotes

[1]We use here the term manifold purely in an informal sense.

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
