[Reviews · NeurIPS 2018]

Reviewer 1



While my concerns were given significant attention in the rebuttal, I feel they were not fully addressed. In particular, regarding comparison with deep ADMM-net and LDAMP, the authors argue that these methods need more training data/training time. However, training time is normally not a big issue (you only train your model once, does it matter if it takes 2 hours or 10?). The *size* of the training data is however important, but no experiments are provided to show superior performance of the proposed method with respect to the the size of training data. This is surprising given that in l. 62. the authors say they use "much less training data" (addressing the challenge of "scarcity of training data" mentioned in l.4 in abstract), without referring back to this claimed contribution anywhere in the paper! Additionally, for the additional super resolution experiments in Table 1 of the rebuttal, I am left again confused. Instead of reporting on a common SR dataset (like one of the datasets SRResnet[12] reports on), they authors report on CelebA and compare with their own implementation of SRResnet on this dataset. The authors do not show a standard baseline, like Bicubic interpolation, meaning the context for these numbers is missing. When looking up in the literature for results on CelebA 8x upscaling (32x32 -> 128x128), (Wavelet-SRNet: A Wavelet-based CNN for Multi-scale Face Super Resolution, Huang et. al., ICCV 2017) gives wildly different numbers. There we see Bicubic having 29.2dB PSNR (higher than the 28.X reported in the rebuttal!), but SSIM of the s.o.t.a. being 0.9432 (lower than the 95.07 SSIM in the rebuttal...). Either the setting is somehow different, or the experiments are somehow flawed, but it is hard to say given that the authors show no standard baselines! On the other side, Reviewer 3 gives a quite different perspective on the paper, focusing on the contraction analysis and the accompanying experiments. While I'm not qualified to comment on the correctness/applicability of this part of the paper, I think Reviewer 3 has valid points that such analysis are valuable in general and should be more often done. ============================================================ The paper proposes an inversion algorithm for ill posed linear measurements of images. That is, from an image x, we are given few (compared to dim(x) ) (noisy) linear measurements from which we want to recover x, relying on the assumption that x lives in some low dimensional manifold. This problem encapsulates for example MRI/CT imaging as well as super-resolution. The proposed approach is an adaptation of the proximal gradient algorithm for deep networks. While I'm not an expert on proximal learning, my understanding is that the authors take the iterative procedure of proximal learning (eq (2)) and swap out the proximal operator (which has closed form solutions for simple priors but otherwise requires solving a convex optimization problem at each iteration), with a deep network. The intuition is that normally the proximal operator can be viewed as taking a step towards the manifold/prior, which is an operation a DNN could in principle learn. This means that instead of deriving the operator from the (known) prior, it needs to be learned from data, motivating the loss (P1) in l. 133. However, this loss is not very clearly explained. Why do we need beta and the (second) measurement consistency term? How does that relate to classical proximal learning? Why not just ask different iterations to also reconstruct the image? How does this term come into the contraction analysis? When the authors jump to the neural proximal training, they essentially just obtain a constrained recursive image restoration network (in the sense that the relationship between x_t and x_t+1 is the composition of a generic DNN and a fixed linear operator g (that depends on the measurement matrix)). Previously recursive approaches for image restorationhave been studied, such as ( Deeply-Recursive Convolutional Network for Image Super-Resolution, Kim et al. CVPR 2016 ) and the authors should discuss the differences. Aside from the fixed g, the training losses differ, but again the loss does not have a clear motivation from proximal learning theory as far as I understand. Experimentally, the paper could also be more convincing. The MRI experiments, while impressive, seem to be done on a private dataset with limited comparisons. Why can't the authors e.g. compare with (Deep ADMM-Net for Compressive Sensing MRI, Sun et al, NIPS 2016) or ( Learned D-AMP: Principled Neural Network Based Compressive Image Recovery, Metzler et al, NIPS 2017)? The Super-Resolution experiments (a field I know well) are extremely limited. The authors claim they only do them to analyse the contraction properties, but do not explain why they need to switch to SR and cannot do this for the MRI experiments. For me, the experiments rather rather raise the question on why the method works so poorly for SR, since this is a problem that falls within the problem formulation (linear measurement matrix). Note I am not saying the authors need to be state of the art in SR, but at least they should report results in a standard manner (e.g. on standard datasets such as Set14, see e.g. Image Super-Resolution Using Dense Skip Connections, Tong et al, ICCV 2017 for a recent s.o.t.a. SR paper). Overall, I think the goal of the paper is worthy (generalizing proximal algorithms to deep net), and the paper is well written in general. My main concern is the execution, both in terms of how the proximal method actually motivates the training objective and the overall experimental evaluation. Minor Comments: l 105: the definition of g is just the argument to the proximal operator in eq (2). I would either define g earlier or refer to (2) in l105. l 129: Shouldn't it be \hat{x}_i = x_i^T instead of x_T^i?

Reviewer 2



This paper proposes a new method of image restoration based on neural proximal gradient descent using a recurrent resnet. The basic idea is to extend the proximal gradient descent algorithm to use a recurrent neural network as the proximal map. Essentially, the recurrent network (they use a ResNet) is used to define the image prior term that, after each update step, moves the estimate towards the image manifold. This paper is well written and contains strong evaluations. However, discussion of related work could be improved, particularly to give an overview for a reader unfamiliar with the problem. Some questions: -Clarity of section 3 could be greatly improved. For example, the superscript/subscript notation appear inconsistent in line 129 and (P1). Also, should an x in the first like of (P1) have a hat? - The consistency term in P1 is noted as being important for convergence. This could be elaborated on for clarity. -Why are the two terms in (P1) weighted by beta and (1-beta), why this inverse relationship? - The method is evaluated on MRI reconstruction and super-resolution. The MRI results appear convincing, outperforming existing methods and proving computationally efficient. - It is unclear how convincing the super-resolution results are. The authors note that super-resolution is a highly studied problem and existing approaches perform very well. Why chose this task to evaluate your approach on then? I'm not sure what is gained from this experiment. This paper falls outside my area of expertise. I am unfamiliar with the problem, the methods, and relevant related work, so this review is honestly just an educated guess.

Reviewer 3



The present submission presents an algorithm and architecture for attacking inverse problems where substantial prior knowledge can be put in the form of a neural network architecture generating the inverses. The idea is simple, modify proximal gradient descent so that the proximal step is a learned neural network, and use a loss that makes the iterates close to the data. The authors also propose an architecture that works well with the algorithm. The paper is marvelously well written (albeit strongly worded, e.g. phrases like 'we develop a successful system solving all these challenges' in the abstract should be toned down unless strong guarantees are provided that address for example the mentioned challenge of data scarcity). The idea is simple, and easy to implement, and the results are very good (given I'm not an expert in any of the datasets). Section 4 is very interesting and well developed, which is arguably of much more use than just this paper since there is growing interest in iterative inference methods. I appreciate that the authors point out the lack of data dependent (distributional) assumptions that ensure approximate convergence with high probability, and I am surprised that the \eta terms are less than one (especially \eta1, which has a large data independent aspect, given the supremum). I would like to know if these terms grow with the number of dimensions of the problem, potentially being larger than 1? This wouldn't be too surprising given the growth of the spectral norms and L1 / LInf norms of random matrices. One of the things that I appreciate the most is section 5. The authors don't just present a contraction result with a potentially meaningless bound in section 4, but study extensively in section 5.3 the behaviour of the contraction analysis by carefully designed experiments that elucidate the properties of the method. More research should be done like this, which is much more informative than crude SNR numbers about the behaviour of the proposed algorithm. Furthermore, numerous ablation studies are performed that help guide how to use the algorithm. I think this is a paper with a clear contribution, a useful simple idea, strong results that are well validated, and perhaps more important than that, the proposed method is examined to the core, with targeted experiments and analysis that are likely to be useful for works to follow. I would like to ask the authors if the code is going to be released with the submission? ========================================================== I think reviewer #1 raises valid concerns regarding the significance of the benchmark results, and the need for more accurate baselines. The reason for my rating was mainly on the experimental work surrounding the workings of the algorithm (which I still stand by), and perhaps my limited knowledge of the datasets lead to an overestimation of the other results' value. I think this is a tricky paper. I still vote for acceptance, given that I think the contraction analysis and related experiments have a strong accompanying value, especially aimed at explaining the algorithm, but nonetheless I think the concerns are valid and the paper should have done a better job regarding the baselines and comparison with previous work. I have thus updated my score to a 7.